# The Clinicopathological Distinction between Immune-Mediated Necrotizing Myopathy and Limb–Girdle Muscular Dystrophy R2: Key Points to Prevent Misdiagnosis

**DOI:** 10.3390/jcm11216566

**Published:** 2022-11-05

**Authors:** Mengge Yang, Suqiong Ji, Li Xu, Qing Zhang, Yue Li, Huajie Gao, Bitao Bu

**Affiliations:** Department of Neurology, Tongji Hospital of Tongji Medical College, Huazhong University of Science and Technology, Wuhan 430030, China

**Keywords:** immune-mediated necrotizing myopathy, limb–girdle muscular dystrophy, clinical features, imaging, histopathology

## Abstract

Background: Limb–girdle muscular dystrophy R2 (LGMD R2) is most frequently misdiagnosed as immune-mediated necrotizing myopathy (IMNM). This study aimed to compare the clinicopathological data of IMNM and LGMD R2 to find distinguishing features. Methods: We retrospectively reassessed the medical data of patients with IMNM (*n* = 41) and LGMD R2 (*n* = 8) treated at Tongji Hospital from January 2017 to December 2021. Results: In our cohort, patients with LGMD R2 had a longer interval of onset to first visit, mild muscle weakness with late upper limb involvement, less myalgia, no cervical muscle weakness or dysphagia, no extramuscular organs affected except cardiac involvement, and lack of various autoantibodies, such as antinuclear antibodies. These features were completely reversed in IMNM. Moreover, thigh MRIs showed that muscle edema prominently affecting the adductor magnus was a characteristic of IMNM, while extensive fatty replacement was more common in LGMD R2 (*p* = 0.0086). Necrotic myofibers presented in both entities (*p* = 0.1693), while features such as ring/whorled and splitting myofibers were more often found in LGMD R2 (*p* = 0.0112 and *p* < 0.0001, respectively). Conversely, sarcoplasmic p62 expression was more pronounced in IMNM (*p* < 0.05). There were 4 of 8 (50%) patients with LGMD R2 initially considered as seronegative IMNM, and therefore unnecessarily treated with immunosuppressive drugs. Insufficient recognition of the early clinical, imaging, and histopathological features of LGMD R2 is the main reason for misdiagnosis. Conclusions: These findings may help clinicians differentiate seronegative IMNM and LGMD R2, reducing early misdiagnosis and mismanagement. Particularly, prominent adductor magnus edema on MRI and abundant p62 staining seem to be good markers for IMNM, while the presence of splitting myofibers is a crucial clue to early hereditary myopathy, including LGMD R2.

## 1. Introduction

Immune-mediated necrotizing myopathy (IMNM) is a recently described subtype of idiopathic inflammatory myopathies (IIMs), which are mainly defined by three subtypes according to different myositis-specific autoantibodies (MSAs), including anti-signal recognition particle (SRP) positive IMNM, anti-3-hydroxy-3-methylglutaryl coenzyme A reductase (HMGCR) positive IMNM, and IMNM with no known autoantibodies (seronegative IMNM) [1]. Patients with IMNM present with rapidly progressive muscle weakness, often requiring early aggressive immunotherapies [1]. Several adult-onset muscular dystrophies can mimic IMNM, leading to clinical misdiagnosis [2,3]. Dysferlinopathy is an autosomal recessively inherited muscular dystrophy caused by mutations in the *DYSF* gene, which encodes the muscle-specific protein dysferlin, and its deficiency leads to sarcolemma repair abnormalities and secondary inflammatory activation [4]. Dysferlinopathy presents with a distal Miyoshi myopathy (MM) phenotype, or more often, a proximal phenotype known as limb–girdle muscular dystrophy R2 (LGMD R2) (formerly LGMD 2B) [5].

With the in-depth understanding of IMNM in recent years, we noted that LGMD R2 is most frequently misdiagnosed as IMNM, especially seronegative IMNM. They have much more similar clinicopathological features, including proximal muscle weakness, remarkably elevated serum creatine kinase (CK), scattered myonecrosis on a muscle biopsy, a relative paucity of inflammatory infiltrates with macrophage predominance, and MAC deposition on the sarcolemma. Genetic testing is useful for distinguishing the two entities; however, most patients with early LGMD R2 lack the typical clinicopathological features. In this scenario, genetic testing is usually considered only in the case of no response to immunotherapies. Antibody testing for HMGCR or SRP is now readily available; however, it plays limited roles in distinguishing seronegative IMNM from LGMD R2. Therefore, the clinicopathological distinction between the two entities is important for early diagnosis and clinical decision making.

Although the features of LGMDR2 and IMNM have been described in previous studies [1,3,6,7], data fully describing the differences between the two entities are still lacking. In this study, we compared the clinicopathological features of IMNM and LGMD R2, investigated the causes of early misdiagnosis of LGMD R2 as seronegative IMNM, and identified some features contributing to the early diagnosis of seronegative IMNM and LGMD R2.

## 2. Materials and Methods

### 2.1. Patients

Muscle biopsies and medical data were obtained from the Department of Neurology at Tongji Hospital from January 2017 to December 2021. The diagnosis for IMNM met the clinico-sero-pathological criteria formulated by the European Neuromuscular Centre International Workshop [8]. The exclusion criteria included drug-, infection-, toxic-, and other MSA-associated myopathies, as well as endocrine myopathy [2,3]. Patients with insufficient medical data were also excluded. A total of 41 patients with IMNM were enrolled, including 23 patients with the anti-SRP autoantibody, 6 patients with the anti-HMGCR autoantibody, and 12 patients with seronegative IMNM. The diagnosis criteria for LGMD R2 are as follows: (1) progressive proximal muscle weakness; (2) pathogenic *DYSF* gene mutation confirmed by genetic analyses; (3) complete or nearly complete dysferlin protein loss (≤ 20%) in muscle biopsies demonstrated by Western blot (WB). A total of 8 patients with LGMD R2 were enrolled. Their pathogenic mutations and WB bands are shown in Appendix A, respectively.

### 2.2. Data Collection

All participants except 2 patients with LGMD R2 were tested for MSAs and myositis-associated autoantibodies (MAAs), including anti-SRP, -HMGCR, -Mi2, -TIF1-γ, -NXP-2, -SAE, -MDA5, -Jo1, -PL7, -PL12, -EJ, -OJ, -Ku, -PMScl100, -PMScl75, and -Ro52 autoantibodies by a commercial laboratory. A line blot immunoassay was used for detection of all MSAs and MAAs except HMGCR, which was performed by a cell-based assay. Connective tissue disease (CTD)-associated autoantibodies, such as antinuclear antibodies, anti-neutrophil cytoplasmic antibodies (ANCA) and rheumatoid factors, serum CK, and lactate dehydrogenase (LDH) levels were routinely performed.

Muscle magnetic resonance imaging (MRI) was performed to assess muscle edema, atrophy, fatty replacement, and myofascial edema. The degree of muscle edema and fatty replacement were graded by the area of muscle involvement as previously described [9]: normal muscle (score 1); <30% of the muscle (score 2); 30–60% of the muscle (score 3); and > 60% of the muscle (score 4). A lung CT scan was performed to detect interstitial lung disease (ILD). Cardiac abnormalities were evaluated by cardiac MRI (myocardial fibrosis, or ventricular/atrial dilatation or dysfunction) and echocardiogram (Echo) (systolic dysfunction or chamber dilatation). Muscle strength was assessed using the Medical Research Council (MRC) grading systems. 

### 2.3. Genetic Analysis

Genetic analysis was performed at the genetic diagnostic center of Tongji Hospital. A whole exome sequencing was tested to identify the pathogenic variants, which was validated using sanger sequencing.

### 2.4. Western Blot

An anti-dysferlin antibody (NCL-hamlet, 1:1000, Leica, Wetzlar, Germany) was used for WB analysis as previously described [10].

### 2.5. Skeletal Muscle Biopsies, and Histological and Immunohistochemical Staining

A muscle biopsy was performed on all patients. Muscle specimens were frozen in isopentane cooled in liquid nitrogen and then stored in a refrigerator at −80 °C. Frozen muscle specimens were sliced into 7 μm sections for histological and immunohistochemical (IHC) staining. Histological staining, such as hematoxylin and eosin (H&E), NADH-tetrazolium reductase (NADH-TR), and acid phosphatase (AcP), was routinely performed. The following primary antibodies were used to recognize: CD4 (1:100, ABclonal, Wuhan, China), CD8 (1:2000, Proteintech, Wuhan, China), CD20, (1:50, ABclonal, Wuhan, China), CD68 (1:100, Santa Cruz Biotechnology, CA, USA), major histocompatibility complex class I (MHC-I, 1:400, Abcam, Cambridge, UK), C5b9 (1:50, DAKO, Copenhagen, Denmark), sequestosome 1 (p62, 1:400, Proteintech, Wuhan, China), and dysferlin (NLC-Hamlet, 1:40, Leica, Wetzlar, Germany).

### 2.6. Histological and IHC Analysis

Five high-power fields (HPFs) for each muscle specimen at 200× magnification were selected for further analysis. Average cell counts (positive staining/HPF) were used for semiquantitative analysis of CD68+ macrophages, CD4+ T cells, CD8+ T cells, and CD20+ B cells. For inflammatory cells, 0 = almost no staining (<5 cells/HPF), 1 = little staining (5–20 cells/HPF), 2 = moderate staining (21–50 cells/HPF), and 3 = abundant staining (> 50 cells/HPF). The comparison of myofibers positive for MAC, MHC-I, and p62 between IMNM and LGMD R2 was determined by the percentage of positive myofibers. For myofibers, 0 = no staining (0% of myofibers), 1 = little staining (< 3% of myofibers), 2 = moderate staining (3–10% of myofibers), 3 = abundant staining (11–30% of myofibers), and 4 = very high staining (> 30% of myofibers). Cell counts were performed using Image J software.

### 2.7. Statistical Analysis

Categorical variables were expressed as frequencies and percentages, while continuous variables were expressed as medians and ranges. Fisher’s exact two-tailed test and the Mann–Whitney U test were used for comparison of categorical variables and continuous variables, respectively. Statistical analysis was performed using GraphPad Prism software. The *p* values < 0.05 were considered statistically significant.

## 3. Results

The clinical and histopathological comparisons between IMNM and LGMD R2 are shown in Table 1 and Table 2, respectively. Patients with IMNM had an older age of onset (*p* = 0.0167) and a shorter interval of onset to first visit (*p* = 0.0175) when compared with patients with LGMD R2. There was no difference in gender between the two conditions.

### 3.1. Initial Presentation

Limb weakness as initial presentation was common in both IMNM and LGMD R2, with proximal muscles especially affected. Early upper and lower limb weakness was seen in most patients with IMNM; however, no upper limb weakness was noted in any of the patients with LGMD R2. One patient (12.5%) with LGMD R2 reported pelvic zone muscle weakness, while some patients with IMNM reported myalgia (19.5%).

### 3.2. Clinical Assessment

At the time of diagnosis, IMNM had more severe muscle weakness (*p* = 0.0046) compared with LGMD R2. Muscle weakness of the upper limbs was more common in IMNM (*p* = 0.0003). Cervical muscle weakness, dysphagia, extramuscular symptoms (skin rash, arthritis, and ILD), and complications (malignancy and CTD) were noted only in IMNM. The frequency of myalgia was markedly increased in IMNM (36.6%), especially in seronegative IMNM (66.67%). Cardiac abnormalities occurred in the two entities. Similar to MSAs, the serum antinuclear antibody was especially common in IMNM (75%), but absent in LGMD R2 (*p* = 0.0003).

### 3.3. Muscle MRI

Representative thigh MRI images of patients with IMNM and LGMD R2 are shown in Figure 1. Muscle edema was more pronounced in IMNM and mainly affected the adductor magnus (AM), followed by posterior thigh muscle groups (long biceps femoris (LB), semitendinosus (ST), and semimembranosus (SM)), the vastus lateralis (VL), and rectus femoris (RF) (Figure 1A,B). Fatty replacement and muscle atrophy were more common in LGMD R2 (*p* = 0.0086, *p* = 0.0277, respectively), and extensive fatty replacement was frequently involved in all three compartments of the thigh (Figure 1C,D).

### 3.4. Histopathology

Histopathological features of IMNM and LGMD R2 are summarized in Table 2, and representative images are shown in Figure 2. Histological analysis showed that myofiber necrosis, regeneration, atrophy, and myophagocytosis presented in both entities (Figure 2A,B,E–H). IMNM trended towards more necrotic myofibers compared to LGMD R2, but was not statistically significant (Figure 2I). Chronic myopathic features, such as internalized nuclei, splitting myofibers, and ring/whorled myofibers were more common in LGMD R2 (*p* = 0.0001, *p* < 0.0001, *p* = 0.0112, respectively). Fiber size variability and fatty replacement were more obvious in LGMD R2, especially in patients with a long course of disease (Figure 2C,D).

A few inflammatory infiltrates were noted in both IMNM and LGMD R2 biopsies with CD68+ macrophage predominance. There were no significant differences in the numbers of CD68+ macrophages, CD4+ T cells, CD8+ T cells, and CD20+ B cells (Appendix A), as well as MHC-I expression and MAC deposition on the sarcolemma between IMNM and LGMD R2. Positive p62 staining was observed in the two entities, whereas the percentage of p62 positive fibers in IMNM biopsies was remarkably higher than in LGMD R2 biopsies (*p* < 0.05, Figure 2M). Dysferlin deficiency only presented in LGMD R2 (Figure 2L); however, dysferlin staining was susceptible to false negative staining (normal IHC labeling) or faint staining (Appendix A).

### 3.5. Diagnosis and Outcome

There were four patients with LGMD R2 (50%) who were initially diagnosed as seronegative IMNM and received immunotherapies. No reliable clues indicative of hereditary myopathy was the leading cause of early misdiagnosis, including no family history (*n* = 4), lack of clinical sign of muscle atrophy (*n* = 3), and no severe fat replacement observed on the muscle biopsy (*n* = 4) or muscle MRI (*n* = 3) (Appendix A). These patients showed no improvement (*n* = 3) or worsened (*n* = 1) after immunotherapies. Then, genetic analysis was performed and confirmed the mutations of the *DYSF* gene.

Most patients with IMNM improved after immunotherapies. Their serum CK dramatically decreased and MRC scores significantly improved (Figure 3A,B). Patients with LGMD R2 progressed slowly, MRC scores decreased with the course of the disease, and serum CK maintained a high level, but had a descending trend (Figure 3A,B).

### 3.6. Seropositive IMNM vs. LGMD R2, and Seronegative IMNM vs. LGMD R2

The clinical and histopathological comparisons between seropositive/seronegative IMNM and LGMD R2 are shown in Table 1 and Table 2, respectively. Overall, the differences were consistent with these findings observed in IMNM and LGMD R2. Of note, myalgia was more common in seronegative patients compared to LGMD R2 patients (*p* = 0.0281).

## 4. Discussion

In our center, up to 50% of patients with LGMD R2 were initially considered as seronegative IMNM, and therefore unnecessarily treated with corticosteroids or immunosuppressors with several possible side effects. We compared detailed clinical, imaging, and histopathological features between IMNM and LGMD R2, and investigated the causes of early misdiagnosis of LGMD R2. Our study highlighted that LGMD R2 can mimic IMNM; however, there were a number of differences between the two entities. Patients with LGMD R2 had a longer interval of onset to first visit, mild muscle weakness with late upper limb involvement, less myalgia, no cervical muscle weakness or dysphagia, no extramuscular organs affected except cardiac involvement, and lack of various autoantibodies including the antinuclear antibody. These clinical features were completely reversed in IMNM. Thigh MRIs showed muscle edema with the adductor magnus prominently affected as a feature of IMNM, while extensive fatty replacement was more common in LGMD R2. Ring/whorled and splitting myofibers were more often found in LGMD R2, while sarcoplasmic p62 expression was more pronounced in IMNM. Insufficient recognition of the early clinical, imaging, and histopathological features of LGMD R2 was the leading cause of misdiagnosis.

IMNM and LGMD R2 frequently presented with proximal weakness; however, the patterns of weakness are remarkably different. Patients with IMNM usually had a proximal weakness with nearly equal involvement of the upper and lower limbs at onset. It then became increasingly apparent that proximal muscle weakness predominantly affected the lower limbs. Patients with LGMD R2 usually started with a proximal muscle weakness of the lower limbs or the pelvic girdle. The upper limbs and shoulder girdle muscles were usually involved later and more mildly during the course of the disease [11].

IMNM is a severe myopathy with an acute or subacute course [12], whereas muscle weakness in patients with LGMD R2 was usually mild at first visit, manifested as difficulty in climbing stairs/mountains, difficulty in rising from the floor, and tiring easily. Mild weakness with significantly elevated CK levels may be a feature of LGMD R2, as there is no relationship between the severity of LGMD R2 and serum CK [10]. Overall, muscle weakness in LGMD R2 progresses slowly with the disease duration, whereas serum CK tends to decrease [10,13].

Myalgia appears to be a clinical feature helping differentiate the two entities. A previous review by Fanin et al. highlighted myalgia as a frequently observed feature of LGMD R2 [5]. However, the largest international multicenter study on dysferlinopathy (including 193 patients) described less common muscle pain (13%) [11]. This is consistent with our result (12.5%) and another multicenter study (5/40) [10]. Muscle atrophy plays a limited role in the early diagnosis of LGMD R2, as mild muscle wasting is difficult to find clinically.

Myocardial abnormalities were frequently detected in both IMNM and LGMD R2, but no patient reported any cardiac symptoms. Cardiac involvement is a relatively common extramuscular manifestation of IMNM [14], associated with a poor prognosis of IMNM [15,16]. The dysferlin protein also presents in the myocardium, and its deficiency leads to myocardial injury [17,18,19]. In a previous study, myocardial damage was detected by cardiac MRI in four of nine patients with LGMD 2B [18]. Symptomatic cardiomyopathy seems to be limited, whereas subclinical cardiac damage does occur in the two entities. Long-term studies are still needed to determine the prognostic significance of subclinical cardiac damage in these patients.

Muscle edema is closely associated with inflammatory activity, suggesting the diagnosis of IIMs, such as IMNM, while extensive fatty replacement is more likely to be an inherited myopathy, such as LGMD R2 [20,21]. However, only one of four patients with LGMD R2 initially considered as IMNM had prominent fat replacement on the thigh MRI (Appendix A). Adductor magnus edema was evident in most IMNM, but usually absent in LGMD R2, suggesting it could be a potentially useful tool to differentiate IMNM from early LGMD R2 or hereditary myopathies.

LGMD featured with variable degrees of dystrophic pathology, including remarkable fiber size variability, increased internalized nuclei, splitting and ring/whorled myofibers, and severe fat replacement, especially in those with a long course of disease [22]. All four patients initially suspected as IMNM lacked severe fat replacement or fiber size variability, whereas splitting myofibers were usually noted (Appendix A), strongly suggesting that splitting myofibers may be a key clue to differentiating early LGMD from IMNM.

Dystrophic pathology was found on the muscle biopsy of one patient with the anti-SRP antibody in our study and several patients with the anti-HMGCR antibody in previous studies [23,24,25]. These patients had a chronic course, resembling LGMD, but responded favorably to immunotherapies [23]. The chronic LGMD-like phenotype appears to be a feature of seropositive IMNM, as there have been no reports of seronegative patients presenting with a slowly progressive weakness. Therefore, serum MSAs should be routinely assessed when patients with suspected LGMD have no family history, and no definitive genetic abnormality or unavailable genetic sequencing. Other serum antibodies, such as the antinuclear antibody, can also be detected in most IMNM, supporting the view that the immune mechanism plays a key role in the pathogenesis of IMNM, but is a secondary, non-specific change in LGMD R2 [4,26].

Immunostaining for p62 (an autophagy marker) may be helpful for distinguishing the two entities. Autophagy is required for successful differentiation of myoblasts and functional regeneration of skeletal muscle [27,28]. Dysferlin deficiency in LGMD R2 leads to damaged muscle regeneration [29,30], which may be responsible for its rarely p62 staining in LGMD R2. Inversely, LC3 (another autophagy marker) and p62 accumulated in regenerating myofibers in IMNM [31], suggesting a better tissue repair ability. Although p62 staining has been considered an interesting diagnostic hallmark of IMNM [1], precise mechanisms of autophagy in IMNM are yet to be elucidated, and much more studies are still needed.

Immunostaining for dysferlin is necessary, but not always reliable in distinguishing the two entities. Almost normal IHC staining was noted in the muscle biopsy from an LGMD R2 patient who had a family history and a pathogenic DYSF mutation. Prior studies have demonstrated that partial dysferlin defects were difficult to identify using IHC staining, leading to false negative staining (normal IHC labeling) or faint staining [5,32]. Harris et al. also observed patients with DYSF mutations presenting with normal dysferlin protein levels [11]. Therefore, gene sequencing is ultimately required to confirm the diagnosis of LGMD R2.

One patient with LGMD R2 progressed gradually during corticosteroid treatment. Her weakness of the lower limbs worsened, followed by upper limb involvement and myalgia. Subsequent genetic analysis confirmed pathogenic mutations in the *DYSF* gene. Then, she was weaned off immunosuppressants and her weakness improved slightly, and myalgia disappeared. Indeed, patients with LGMD R2 seem to worsen and develop a series of side effects after corticosteroid or immunosuppressive treatment [5,33,34]. Our study provides a toolkit for clinicians to timely screen possible LGMD R2 from IMNM, especially seronegative IMNM, avoiding the use of potentially harmful immunosuppressive drugs.

However, there are still some limitations. First, some muscular dystrophy patients may be hidden in seronegative cases. Considering the possibility of a mistaken diagnosis of seronegative IMNM, we made a comparison between seropositive IMNM and LGMD R2 at the same time. Second, other recessive LGMDs and distal MM phenotypes were not included in our study. We focused on the differences between IMNM and LGMD R2 rather than all LGMDs, as LGMD R2 is the most common subtype in our country, accounting for 49.5% of all LGMDs [35]. Moreover, as shown, the clinicopathological features of LGMD R2 are more similar to IMNM. Particularly, sarcolemmal MAC deposition can be observed in LGMD R2, but is rarely seen in other muscular dystrophies [8,36]. Distal MM features initial distal weakness and prominent calf muscle atrophy, particularly the gastrocnemius involvement, which is not often misdiagnosed as IMNM [37]. One patient with distal MM was excluded from our study due to incomplete medical data. Even so, most findings are suitable for distal MM, as there is no striking difference in clinical presentation (except weakness pattern) and examination between distal MM and LGMD R2 [11]. Finally, this study was limited by a small number of patients with LGMD R2, leading to potential information scarcity and selection bias, such as the lack of patients with asymptomatic hyperCKemia at onset.

## 5. Conclusions

Our study provided some important differences between the two entities, which may help clinicians timely screen possible LGMD R2 from seronegative IMNM for genetic testing, thereby shortening the time to diagnosis. No reliable clues indicative of hereditary myopathy was the leading cause of early misdiagnosis of LGMD R2 as seronegative IMNM. In this scenario, we highlighted that abundant p62 staining and muscle edema affecting the adductor magnus on MRI seem to be good markers for IMNM, while the presence of splitting myofibers is a crucial clue to early hereditary myopathy, including LGMD R2. It is important to consider a combination of clinicopathological features in any case. Prospective and multi-center studies with a larger sample of patients are still needed to confirm our preliminary observations and explore the differences between IMNM and others’ hereditary myopathy, especially LGMD.

## Figures and Tables

**Figure 1 jcm-11-06566-f001:**
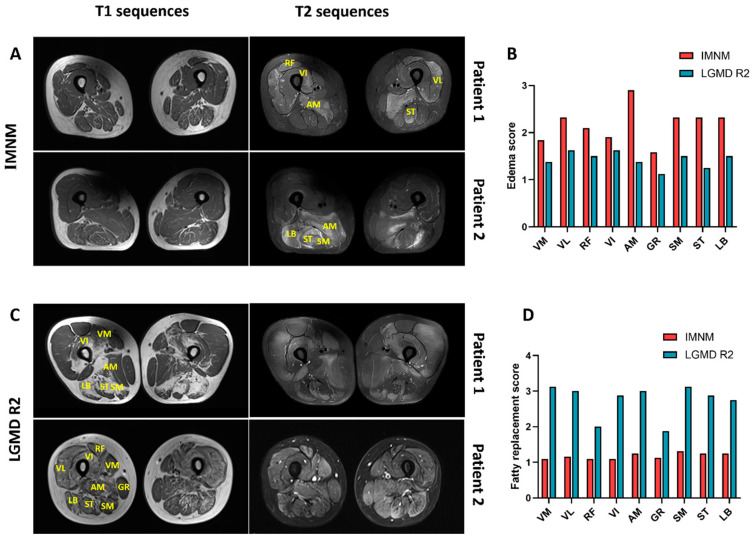
Representative thigh muscle MRI of IMNM and LGMD R2. (**A**) Slight or no fatty replacement on T1 images and a marked T2 hyperintensity in patients with IMNM. (**B**) Muscle edema with adductor magnus (AM) prominently affected. (**C**) Extensive fatty replacement on T1 images and slight T2 hyperintensity in patients with LGMD R2. (**D**) Fatty replacement occurred in all three compartments of the thigh. Note: VM: vastus medialis; VL: vastus lateralis; RF: rectus femoris; VI: vastus internus; GR: gracilis; SM: semimembranosus; ST: semitendinosus; LB: long biceps femoris. IMNM: *n* = 31; LGMD R2: *n* = 8.

**Figure 2 jcm-11-06566-f002:**
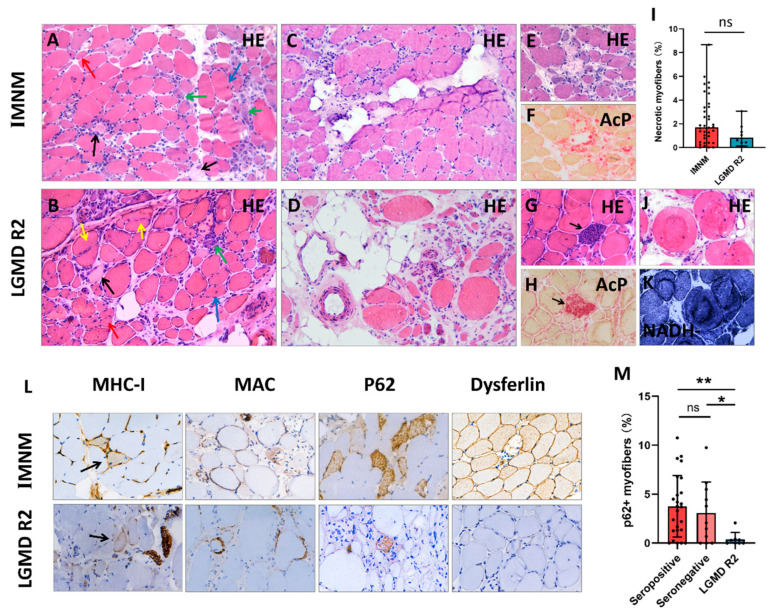
Representative histopathological images of IMNM and LGMD R2. (**A**,**B**) Hematoxylin-eosin (H&E) staining showing profound necrotic myofibers (black arrows), regenerating myofibers (basophilic cells, green arrows), degenerating myofibers (myofibers with internalized nuclei, blue arrows), atrophic myofibers (red arrows), and splitting myofibers (yellow arrows). (**C**) Slight fat replacement in an IMNM biopsy. (**D**) Severe fat replacement with fiber size variability (dramatically atrophic or enlarged myofibers) in an LGMD R2 patient with a long course of disease. (**E**–**H**) H&E and acid phosphatase (AcP) staining showing necrotic myofibers undergoing myophagocytosis. (**I**) More necrotic myofibers in IMNM biopsies. (**J**,**K**) H&E and NADH staining showing ring/whorled myofibers in a LGMD R2 biopsy. (**L**) Immunohistochemical staining for major histocompatibility complex class I (MHC-I, black arrows), membrane attack complex (MAC), sequestosome 1 (p62), and dysferlin. (**M**) More myofibers positive for p62 in seropositive and seronegative IMNM biopsies. Images (**A**–**D**): original magnification 200 ×; images (**E**–**H**,**J**–**L**): original magnification 400×. ns, no significance; * *p* < 0.05, ** *p* < 0.01.

**Figure 3 jcm-11-06566-f003:**
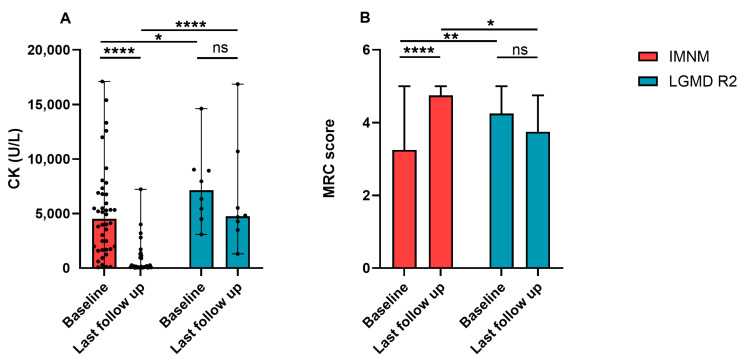
Changes of serum CK levels and MRC scores of patients with IMNM and LGMD R2. (**A**) CK levels of patients with IMNM and LGMD R2 at baseline and the last follow-up. (**B**) MRC scores of patients with IMNM and LGMD R2 at baseline and the last follow-up. Data was shown by medians and range. CK: creatine kinase; MRC: Medical Research Council; ns: no significance; * *p* < 0.05, ** *p* < 0.01, **** *p* < 0.0001.

**Table 1 jcm-11-06566-t001:** Comparison of the clinical features between IMNM and LGMD R2.

Clinical Features	All IMNM (*n* = 41)	Seropositive (*n* = 29)	Seronegative (*n* = 12)	LGMD R2 (*n* = 8)	*p*1 Value	*p*2 Value	*p*2 Value
**Age of onset (years)**	48 (11–73)	48 (11–65)	46 (24–73)	30.5 (24–57)	**0.0167**	**0.0325**	**0.0209**
**Female** (*n*, %)	27 (65.8)	20 (68.97)	7 (58.3)	3 (37.5)	0.2367	0.2152	0.6499
**Onset to first visit** (months)	3 (0.1–24)	4 (1–24)	1.5 (0.1–12)	18 (1–60)	**0.0175**	**0.0396**	**0.0143**
**Initial symptom**							
Muscle weakness	31 (75.6)	23 (79.31)	8 (66.67)	8 (100)	0.3292	0.3053	0.1107
Proximal dominant	30 (73.1)	21 (72.41)	8 (66.67)	8 (100)	0.1718	0.1598	0.1107
Lower limbs	30 (73.1)	23 (79.31)	7 (58.33)	7 (87.5)	0.6598	>0.9999	0.3246
Upper limbs	22 (53.6)	16 (55.17)	6 (50)	0 (0)	**0.0056**	**0.0056**	**0.0419**
Pelvic strap muscle	0 (0)	0 (0)	0 (0)	1 (12.5)	>0.9999	>0.9999	>0.9999
Myalgia	8 (19.5)	4 (13.79)	4 (33.33)	0 (0)	0.3220	0.5574	0.1166
Other initial symptoms	2 (4.9)	2 (6.9)	0 (0)	0 (0)	>0.9999	>0.9999	>0.9999
**Clinical assessment**							
Muscle strength (MRC)	3.25 (1–5)	3.25 (1–5)	3 (1–5)	4 (4–5)	**0.0012**	**0.0029**	**0.0029**
Severe weakness (MRC ≤ 3)	23 (56.1)	16 (55.17)	7 (58.33)	0 (0)	**0.0046**	**0.0056**	**0.0056**
Lower limbs weakness	41 (100)	29 (100)	12 (100)	8 (100)	>0.9999	>0.9999	>0.9999
Upper limbs weakness	38 (92.7)	26 (89.66)	12 (100)	4 (50)	**0.0003**	**0.0271**	**0.0144**
Lower limbs >upper limbs	31 (75.61)	21 (72.41)	10 (83.3)	7 (87.5)	0.6633	0.6487	>0.9999
Proximal >distal	35 (85.4)	25 (86.2)	10 (83.3)	7 (87.5)	>0.9999	>0.9999	>0.9999
Muscle atrophy	9 (22)	8 (27.59)	1 (8.33)	4 (50)	0.1834	0.2328	0.1089
Myalgia	15 (36.6)	7 (24.14)	8 (66.67)	1 (12.5)	0.2454	0.3945	**0.0281**
Cervical muscle weakness	4 (9.8)	3 (10.34)	1 (8.33)	0 (0)	>0.9999	0.6555	>0.9999
Dysphagia	5 (12.2)	5 (17.24)	0 (0)	0 (0)	0.5751	>0.9999	>0.9999
Dyspnea	0 (0)	0 (0)	0 (0)	0 (0)	>0.9999	>0.9999	>0.9999
**Extramuscular symptom**	16 (39.02)	11 (37.93)	5 (41.67)	0 (0)	**0.0394**	0.0757	0.0547
Skin rash	2 (4.9)	2 (6.9)	0 (0)	0 (0)	>0.9999	>0.9999	>0.9999
Arthritis	2 (4.9)	1 (3.45)	1 (8.33)	0 (0)	>0.9999	>0.9999	>0.9999
ILD	13/35 (37.1)	9/24 (37.5)	4/11 (36.36)	0/6 (0)	0.1523	0.1405	0.2374
**Cardiac involvement**	11/31 (35.5)	7/21 (33.33)	4/10 (40)	3/6 (50)	>0.9999	>0.9999	>0.9999
Evaluated by cardiac MRI	8/16 (50)	6/12 (50)	2/4 (50)	2/2 (100)	--	-	-
Evaluated by Echo	3/32 (9.3)	1/26 (3.85)	2/6 (33.33)	1/6 (16.7)	--	-	-
**Complication**	8 (19.5)	6 (20.69)	2 (16.67)	0 (0)	0.3220	0.3053	0.4975
Malignancy	3 (7.3)	2 (6.9)	1 (8.33)	0 (0)	>0.9999	>0.9999	>0.9999
CTD	5 (12.2)	4 (13.8)	1 (8.33)	0 (0)	0.5751	0.5574	>0.9999
**Laboratory testing**							
CK (U/L)	4111 (64–17,100)	4912 (64–17,100)	4036 (108–15,393)	7138.5 (3084–14611)	**0.0485**	**0.0469**	0.1813
LDH (U/L)	739 (140–2712)	782 (143–2712)	544.5 (140–1867)	498.5 (306–839)	0.1823	0.0785	0.8928
MSAs (anti-SRP or -HMGCR)	29/41 (70.7)	29/29 (100)	0 (0)	0/6 (0)	**0.0019**	**<0.0001**	>0.9999
Antinuclear antibody	30/40 (75)	24/28 (85.71)	6/12 (50)	0/7 (0)	**0.0003**	**<0.0001**	**0.0436**
ANCA	1/33 (3)	1/24 (4.17)	0/9 (0)	0/6 (0)	>0.9999	>0.9999	>0.9999
**Muscle MRI**							
Edema	39/39 (100)	27/27 (100)	12/12 (100)	7/8 (87.5)	>0.9999	>0.9999	>0.9999
Fatty replacement	9/39 (23.08)	5/27 (18.51)	4/12 (33.33)	6/8 (75)	**0.0086**	**0.0058**	0.1698
Atrophy	8/39 (20.5)	7/27 (25.93)	1 (8.33)	5/8 (62.5)	**0.0277**	0.0912	**0.0181**
Fascial edema	2/39 (5.1)	2/27 (7.4)	0 (0)	2/8 (25)	0.1290	0.2179	>0.9999
**Immunological therapy**							
Corticosteroids	37 (90.2)	28 (96.55)	9 (75)	4 (50)	**0.0055**	**0.0014**	0.2031
Immunosuppressant	24 (58.5)	19 (65.51)	5 (41.67)	1 (12.5)	**0.0232**	**0.0140**	0.3246
Other (IVIG or Rituximab)	2 (4.9)	2 (6.9)	0 (0)	0 (0)	>0.9999	>0.9999	>0.9999
**Follow-up**							
Period, months, median (range)	32 (6–55)	30 (3–48)	23 (4–55)	12.5 (11–51)	0.5245	0.2403	0.7199
Lost to follow-up	3 (7.3)	2 (6.9)	1 (8.33)	0 (0)	>0.9999	>0.9999	>0.9999
MRC score at the last follow-up	4.75 (3–5)	4.75 (3–5)	4.75 (3–5)	3.75 (3–4.75)	**0.0110**	**<0.0001**	**0.0382**
CK at the last follow-up (U/L)	150 (25–7225)	225 (38–4000)	100 (25–7225)	4680 (1300–16855)	**<0.0001**	**<0.0001**	**0.0025**
Clinical improvement	38/38 (100)	29/29 (100)	11/11(100)	0 (0)	**<0.0001**	**<0.0001**	**<0.0001**

Abbreviation: ANCA: Anti-neutrophil cytoplasmic antibodies; CTD: connective tissue diseases; CK: creatine kinase; Echo: echocardiogram; IMNM: immune-mediated necrotizing myopathy; ILD: interstitial lung disease; IVIG: intravenous immunoglobulin; LDH: lactate dehydrogenase; LGMD R2: Limb–band muscular dystrophy R2; MRC: Medical Research Council; MSAs: myositis-specific autoantibodies; MRI: magnetic resonance imaging. Note: Clinical assessment was performed at time of diagnosis. Categorical variables were expressed as frequencies and percentages (*n* (%)), while continuous variables were expressed as median and range. p1: All IMNM vs. LGMD R2; p2: seropositive IMNM vs. LGMD R2; p3: seronegative IMNM vs. LGMD R2.

**Table 2 jcm-11-06566-t002:** Comparison of the histopathological features between IMNM and LGMD R2.

Pathological Features	IMNM (*n* = 41)	Seropositive (*n* = 29)	Seronegative (*n* = 12)	LGMD R2 (*n* = 8)	*p*1 Value	*p*2 Value	*p*3 Value
HE staining	*n* = 41	*n* = 29	*n* = 12	*n* = 8			
Myonecrosis	40 (97.6)	28 (96.55)	12 (100)	8 (100)	>0.9999	>0.9999	>0.9999
Regeneration	36 (87.8)	26 (89.66)	10 (83.3)	8 (100)	0.5751	>0.9999	0.4947
Internalized nuclei (≥ 10%)	6 (14.6)	6 (20.69)	0 (0)	7 (87.5)	**0.0001**	**0.0011**	**0.0001**
Splitting myofibers	1 (2.4)	1 (3.4)	0 (0)	7 (87.5)	**<0.0001**	**<0.0001**	**0.0001**
Ring or whorled myofibers	1 (2.4)	1 (3.4)	0 (0)	3 (37.5)	**0.0112**	**0.0256**	**0.0491**
Fatty replacement	10 (24.4)	7 (24.13)	3 (33.33)	4 (50)	0.0815	0.2035	0.3563
IHC for inflammatory cells	*n* = 34	*n* = 23	*n* = 11	*n* = 8			
CD68+ macrophage	32 (94.1)	22 (96.65)	10 (90.9)	8 (100)	>0.9999	>0.9999	>0.9999
CD4+ helper T cells	31 (91.2)	21 (91.3)	10 (90.9)	6 (75)	0.2368	0.2683	0.5459
CD8+ cytotoxic T cells	21 (61.8)	16 (69.56)	5 (45.4)	6 (75)	0.6888	>0.9999	0.3521
CD20+ B cells	6 (17.6)	6 (26.09)	0 (0)	1 (12.5)	>0.9999	0.6417	0.4211
IHC analysis	*n* = 34–36	*n* = 23–25	*n* = 11	*n* = 8			
MHC-I expression on sarcolemma	20/34 (58.8)	14/23 (60.87)	6 (54.5)	4 (50)	0.7061	0.6894	>0.9999
MAC deposition on sarcolemma	14/34 (41.2)	10/23 (43.48)	4 (36.4)	1 (12.5)	0.2225	0.2028	0.3378
p62 expression on sarcoplasm	27/36 (75)	21/25 (84.0)	8 (72.7)	4 (50)	0.2089	0.0737	0.3765
Dysferlin deficiency	0 (0)	0 (0)	0 (0)	7 (87.5)	**<0.0001**	**<0.0001**	**<0.0001**

Abbreviation: HE: hematoxylin–eosin; IHC: immunohistochemistry; IMNM: immune-mediated necrotizing myopathy; LGMD R2: Limb–band muscular dystrophy R2; MHC-I: major histocompatibility complex class I; MAC: membrane attack complex. Note: Data were expressed as frequencies and percentages (*n* (%)). The degree of internalized nuclei was shown by the percentage of myofibers with internalized nuclei. 0 = no internalized nuclei (<3%); 1 = scarce internalized nuclei (3–10%); 2 = more internalized nuclei (10–50%); 3 = abundant internalized nuclei (≥ 50%). p1: All IMNM vs. LGMD R2; p2: seropositive IMNM vs. LGMD R2; p3: seronegative IMNM vs. LGMD R2.

## Data Availability

The data presented in this study are available in the article or Appendix A.

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
