# Peer review of "The Clinicopathological Distinction between Immune-Mediated Necrotizing Myopathy and Limb–Girdle Muscular Dystrophy R2: Key Points to Prevent Misdiagnosis"

_jcm, 2022, doi:10.3390/jcm11216566_

Round 1
Reviewer 1 Report
I have only minor revisions:
1. line 87: specify commercial laboratory methods (i.e. lineblot immunoassay for all myositis-associated antibodies except HMGCR, that is done by ELISA I think or CLIA)
2. line 88: ANA now is the acronymus for anti-nucleocytoplasmic antibodies, including also all myosistis-associaed antibodies which present commonly cytoplasmic staining on HEp2
3. table 1 and 2: some numerical data are disaligned with features in the first column, please revised
4. all paper: why sometimes words or entire part are underlined? please correct
Reviewer 2 Report
The article of Yang and Colleagues is interesting and useful, well written and complete
I suggest revising the syllable synonym division, and the unnecessary capital letter use for Whole, row102.
In Table 1 would be helpful to evidence significant p values (as shown in Table 2); please put abbreviations in alphabetic order in Table 1.
Please modify the unnecessary underlined sentences on pages 10-11
